# Prioritizing interventions for preventing COVID-19 outbreaks in military basic training

**Guido España**[1], **T. Alex Perkins**[1]*, **Simon D. Pollett**[2], **Morgan E. Smith**[1], **Sean M. Moore**[1], **Paul O. Kwon**[3], **Tara L. Hall**[3], **Milford H. Beagle, Jr.**[4], **Clinton K. Murray**[5], **Shilpa Hakre**[6,7], **Sheila A. Peel**[8], **Kayvon Modjarrad**[6], **Paul T. Scott**[6]*

1 Department of Biological Sciences and Eck Institute for Global Health, University of Notre Dame, Notre Dame, Indiana, United States of America, 2 Viral Diseases Branch, Walter Reed Army Institute of Research, Silver Spring, Maryland, United States of America, 3 Moncrief Army Health Clinic, Columbia, South Carolina, United States of America, 4 United States Army Training Center, Fort Jackson, Columbia, South Carolina, United States of America, 5 Walter Reed Army Institute of Research, Silver Spring, Maryland, United States of America, 6 Emerging Infectious Diseases Branch, Walter Reed Army Institute of Research, Silver Spring, Maryland, United States of America, 7 Henry M. Jackson Foundation for the Advancement of Military Medicine, Inc., Bethesda, Maryland, United States of America, 8 Diagnostics and Countermeasures Branch, Walter Reed Army Institute of Research, Silver Spring, Maryland, United States of America

☙ These authors contributed equally to this work.

* taperkins@nd.edu (TAP); paul.t.scott.civ@mail.mil (PTS)

**Data Availability Statement:** All data and code are available in a repository titled "Prioritizing interventions for preventing COVID-19 outbreaks in military basic training" on GitHub, which can be

## Abstract

Like other congregate living settings, military basic training has been subject to outbreaks of COVID-19. We sought to identify improved strategies for preventing outbreaks in this setting using an agent-based model of a hypothetical cohort of trainees on a U.S. Army post. Our analysis revealed unique aspects of basic training that require customized approaches to outbreak prevention, which draws attention to the possibility that customized approaches may be necessary in other settings, too. In particular, we showed that introductions by trainers and support staff may be a major vulnerability, given that those individuals remain at risk of community exposure throughout the training period. We also found that increased testing of trainees upon arrival could actually increase the risk of outbreaks, given the potential for false-positive test results to lead to susceptible individuals becoming infected in group isolation and seeding outbreaks in training units upon release. Until an effective transmission-blocking vaccine is adopted at high coverage by individuals involved with basic training, need will persist for non-pharmaceutical interventions to prevent outbreaks in military basic training. Ongoing uncertainties about virus variants and breakthrough infections necessitate continued vigilance in this setting, even as vaccination coverage increases.

## Author summary

COVID-19 has presented enormous disruptions to society. Militaries are not immune to these disruptions, with outbreaks in those settings posing threats to national security. We present a simulation model of COVID-19 outbreaks in a U.S. Army basic training setting to inform improved approaches to prevention there. Counterintuitively, we found that outbreak risk is driven more by virus introductions from trainers than the large number

accessed at https://github.com/confunguido/
prioritizing_interventions_basic_training.

**Funding:** This work was supported by the U.S.
Department of Defense (DoD) Defense Health
Program (DHP) Research, Development, Test, and
Evaluation (RDT&E) funds appropriated to the
Walter Reed Army Institute of Research (WRAIR)
and was executed through a contract (No.
W81XWH20C0072) between the U.S. Army
Medical Research and Development Command and
the University of Notre Dame (TAP). The funders
had no role in study design, data collection and
analysis, decision to publish, or preparation of the
manuscript.

**Competing interests:** The authors have declared
that no competing interests exist.

of trainees, and that outbreak risk is highly sensitive to false-positive results during entry testing. These findings suggest practical ways to improve prevention of COVID-19 outbreaks in basic training and, as a result, maintain the flow of new soldiers into the military. This work highlights the need for bespoke modeling to inform prevention in diverse institutional settings.

## Introduction

In addition to the widespread societal and economic harms caused by the COVID-19 pandemic, operations in numerous institutional settings have experienced disruptions and necessitated major adjustments. As one example, colleges and universities have adopted a variety of testing strategies to reduce transmission, including pre-matriculation testing and up to twice-weekly testing to survey for asymptomatic and pre-symptomatic infections [1–3]. Some sporting leagues and workplaces have made similar adjustments to their operations [4, 5]. In situations where adjustments have been more minimal, such as relying solely on symptom-based surveillance, large outbreaks have occurred, requiring the suspension of operations until the outbreak has run its course [6–9]. These failures indicate that symptom-based surveillance is inadequate as the primary intervention for preventing the introduction and spread of SARS-CoV-2 in these settings [10–13].

Beyond the aforementioned institutional settings, COVID-19 has also caused disruption in military settings [14–16]. On the USS Theodore Roosevelt, an outbreak of COVID-19 infected at least 1,331 out of 4,779 sailors and forced the diversion of the ship to the U.S. Naval base on Guam [14,15]. In several basic training settings, COVID-19 outbreaks have occurred shortly after trainees arrived, despite the fact that they were tested on arrival and isolated if positive [16,17]. Outbreaks in basic training settings are of concern because they disrupt the flow of new soldiers into the military, which is essential to maintaining force strength as retirements and expiring enlistments continue despite interruptions to basic training.

The fact that outbreaks have occurred in basic training despite efforts to prevent them suggests that there is room for improvement with prevention in this setting [18,19]. There are a number of unique challenges to preventing outbreaks during basic training, however. First, new recruits to the military are generally in good health and young, making it likely that they develop only mild symptoms or none at all [20,21]. Second, basic training involves groups of hundreds of people training in close quarters (e.g. first aid, partner-based strength and conditioning) and spending nearly all of their time together for 70 days, including in situations that present prime opportunities for transmission, such as dining, sleeping, exercising, and performing personal hygiene. Third, reverse-transcriptase polymerase chain reaction (RT-PCR) testing at the time of arrival leaves open the possibility of missing infections among trainees who were infected shortly before arrival or en route [22,23]. In addition, the current lack of regular testing of trainers and support staff leaves open the possibility that they could introduce the virus into this setting.

We used an agent-based simulation model developed around a hypothetical basic training setting (Fig 1) to investigate the potential to reduce the risk and extent of COVID-19 outbreaks in this unique setting. We calibrated the model to data from testing upon arrival and 18–22 days later at two U.S. Army posts that experienced COVID-19 outbreaks. The calibration informed the model's assumptions about the initial prevalence of infection among recruits and transmission potential in a basic training setting, as represented by the basic reproduction number, $R_0$. Using this model, we examined how effective four interventions might be in

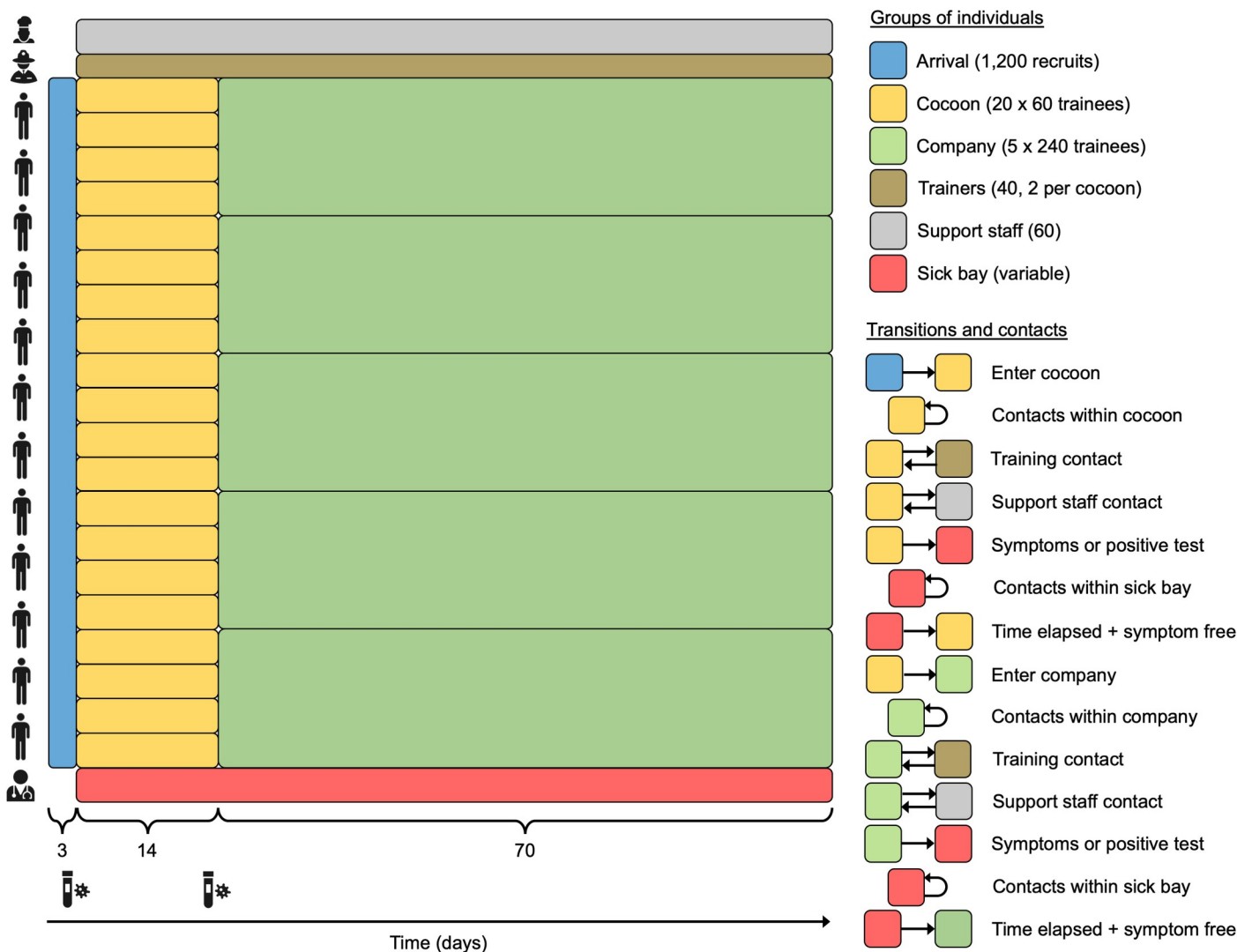

**Fig 1. Model schematic.** Trainees arrive in a three-day window (blue), progress to cocoons of 60 trainees each for 14 days (yellow), and then progress to companies of 240 trainees each for 56 days (green). Trainees have contact with other trainees in their cocoon or company, with trainers (brown) assigned to their unit (two per cocoon, eight per company), and with support staff (gray). Trainees who test positive following arrival testing or presentation with symptoms are placed in the sick bay (red) for ten days before returning to their unit. Trainers and support staff who test positive following presentation with symptoms isolate from home for ten days. New cohorts like the one depicted here enter training posts on a weekly basis, but we model only one given that cohorts do not interact with one another. All processes in the model are defined on a daily time step.

reducing the probability and size of outbreaks in this setting: 1) reducing introductions of the virus into the basic training setting by trainers and support staff, 2) increasing rounds of arrival testing of trainees, 3) increasing compliance with wearing face masks and practicing physical distancing, and 4) increasing immunity among trainees through pre-arrival vaccination.

## Results

### Model calibration

We assumed values of most parameters based on the literature (Table 1) and calibrated two others (initial prevalence of infection, $p$, and the basic reproduction number, $R_0$) for each of two U.S. Army posts with known outbreaks during basic training: Fort Benning (FB) and Fort

**Table 1. Model parameters.** For parameters for which references are cited, baseline values correspond to median estimates, and low and high values correspond to 2.5% and 97.5% quantiles, respectively. Isolation length is the only exception, which was chosen to span a range of values that have been considered at different points.

| Parameter | Baseline value | Low value | High value | Reference |
|---|---|---|---|---|
| *Parameters calibrated to outbreak data* | | | | |
| Basic reproduction number | 5.0 | 4.0 | 6.0 | Calibrated, +/- 1 |
| Initial prevalence of infection | 0.014 | 0.0025 | 0.022 | Calibrated |
| *Parameters explored in intervention analysis* | | | | |
| Probability of community exposure to trainers and support staff over the 70 days of basic training | 0.01 | 0 | 0.10 | Pei et al. [24] |
| Compliance with masks and distancing | 0.3 | 0.1 | 0.5 | Chu et al. [25] |
| Proportion immune upon arrival | 0.026 | 0.018 | 0.033 | Pei et al. [24] |
| *Parameters explored in sensitivity analysis* | | | | |
| Incubation period (shape) | 5.807 | 3.585 | 13.865 | Lauer et al. [22] |
| Incubation period (scale) | 0.948 | 0.368 | 1.696 | Lauer et al. [22] |
| Duration of symptoms | 10 d | 8 d | 11 d | Chen et al. [26] |
| Proportion symptomatic | 0.57 | 0.54 | 0.60 | Kasper et al. [14] |
| Generation interval (shape) | 2.89 | 1.7 | 4.7 | Ferretti et al. [27] |
| Generation interval (scale) | 5.67 | 4.6 | 6.9 | Ferretti et al. [27] |
| Test specificity | 0.998 | 0.992 | 0.999 | Perkins et al. [28] |
| Test sensitivity | 0.859 | 0.547 | 0.994 | Perkins et al. [28] |
| Protection from face masks (odds ratio) | 0.3 | 0.2 | 0.5 | Payne et al. [15] |
| Isolation length | 10 d | 7 d | 14 d | CDC [29] |
| Relative infectiousness of asymptomatics | 0.8 | 0.5 | 1.0 | Assumed |

Leonard Wood (FLW). Upon arrival, 4/640 recruits were positive at FB and 0/500 at FLW. After accounting for the possibility of false negatives and false positives consistent with our model's assumptions about test sensitivity and specificity, we obtained median estimates of $p$ of 1.9% at FB (95% credible interval: 0.3–2.9%) and 0.9% at FLW (95% CrI:0.2–1.5%). Simulating the model forward until the next testing day on each post, we found that a median $R_0$ value of 5.0 (95% CrI: 3.9–6.1) best matched the 142/636 positive tests on day 22 at FB and that a median $R_0$ value of 4.9 (95% CrI: 3.9–6.2) best matched the 70/500 positive tests on day 18 at FLW (Figs 2 and S1). Given that these outbreaks were exceptional events rather than common occurrences, we focused our baseline scenario on a value of $R_0$ equal to the average of the 0.1% quantiles of the $R_0$ estimates from FB and FLW (3.4).

## Model behavior under baseline scenario

Following the calibration procedure, simulations of our model tracked a cohort of 1,200 trainees who spent their first two weeks of basic training in 20 cocoons of 60 and the next eight weeks in five companies of 240 (Fig 1). Under the baseline scenario, testing occurred upon arrival and 14 days later, compliance with face masks and physical distancing was assumed as 30%, the proportion immune at the time of arrival was 2.6%, and trainers and support staff had a 1% chance of becoming infected in the community over the 70-day period of basic training. Given that calibrated values of $p$ and $R_0$ from FB and FLW were similar, we used the averages of their medians for $p$ (1.3%) and 0.1% quantiles for $R_0$ (6.1) in our baseline scenario. We chose the 0.1% quantile for $R_0$ because the outbreaks at FB and FLW were unusually large and because values of $R_0$ within this range resulted in model behavior with greater sensitivity to changes in other model parameters than higher values of $R_0$ did.

Under this scenario, cumulative infections over the 70-day training period were distributed multimodally across 1,000 replicate simulations, with many resulting in very few infections

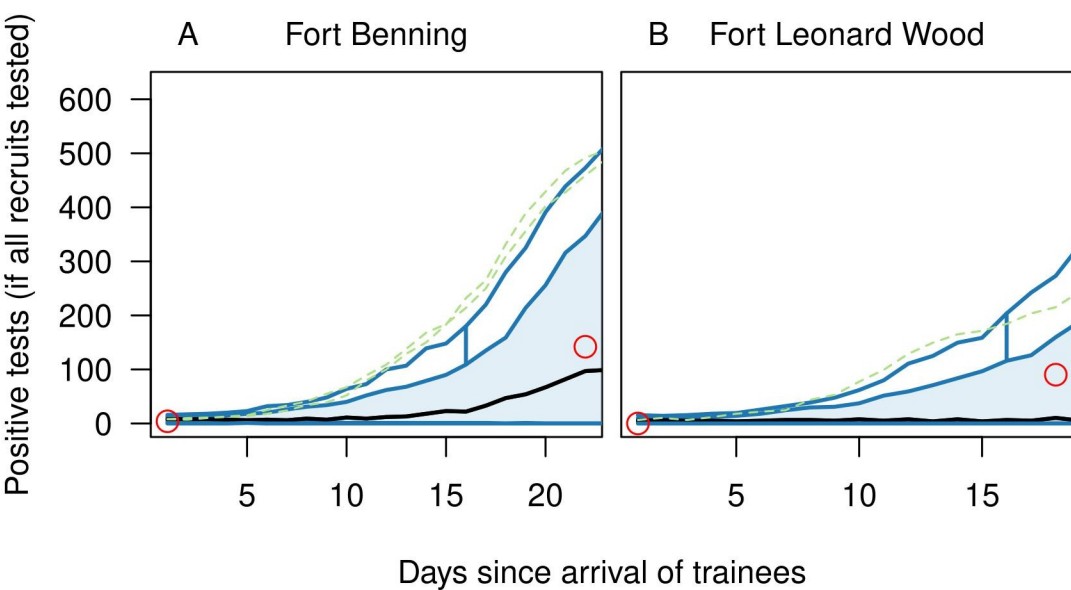

**Fig 2. Model calibration to data from outbreaks in basic training settings.** Functional boxplots show model predictions from 1,000 replicate simulations for Fort Benning (left) and Fort Leonard Wood (right) based on median parameter values calibrated to data from each (red circles). The functional boxplot shows the median estimate (black line), 50% central region (25–75%) (blue band), 1.5 times the central region (blue lines), and outliers defined as lines outside of 1.5 times the central region (dashed green lines). These results demonstrate agreement between the data and the central tendency of the model but also highlight the degree of stochasticity in the model's behavior.

and some resulting in outbreaks in one or more companies (Fig 3A). This multimodal pattern was driven by stochasticity in the number of companies that experienced an outbreak affecting many individuals within the company but few outside it, consistent with the structure of contacts assumed in the model. Based on this distribution, we defined an outbreak as a simulation in which 100 or more infections occurred over the training period. According to this definition (which we refer to as an outbreak hereafter), 71% of simulations resulted in an outbreak, and the median size of an outbreak was 427 (25–75% interval: 220–639) in the event that one occurred (Fig 3B). When all interventions in the baseline scenario were relaxed (i.e., no arrival testing, no symptom-based surveillance and isolation, and no masks or distancing), the probability of an outbreak increased to 0.95, and the median size of an outbreak increased to 663 (25–75% interval: 444–882) (Fig 3C and 3D). Thus, our model predicts that even though interventions under our baseline scenario allowed for outbreaks, they made them less frequent and less severe than they would have been otherwise. Still, results from the baseline scenario indicate that there is scope for further reducing outbreak risk.

## Impact of interventions

**Reducing introductions by trainers and support staff.** When introductions by trainers and support staff were eliminated completely, the probability of an outbreak decreased from 0.73 to 0.48 under our model (Fig 4). The size of outbreaks was also reduced, with median cumulative infections decreasing from 428 to 406 and the 75th quantile decreasing from 636 to 435. This pattern reflects a decrease in the number of companies experiencing an outbreak, consistent with the multimodal nature of how cumulative infections were distributed across replicate simulations (Fig 3A and 3C). When the probability of community exposure for trainers and support staff increased from 0.01 to 0.10 over the course of the training period,

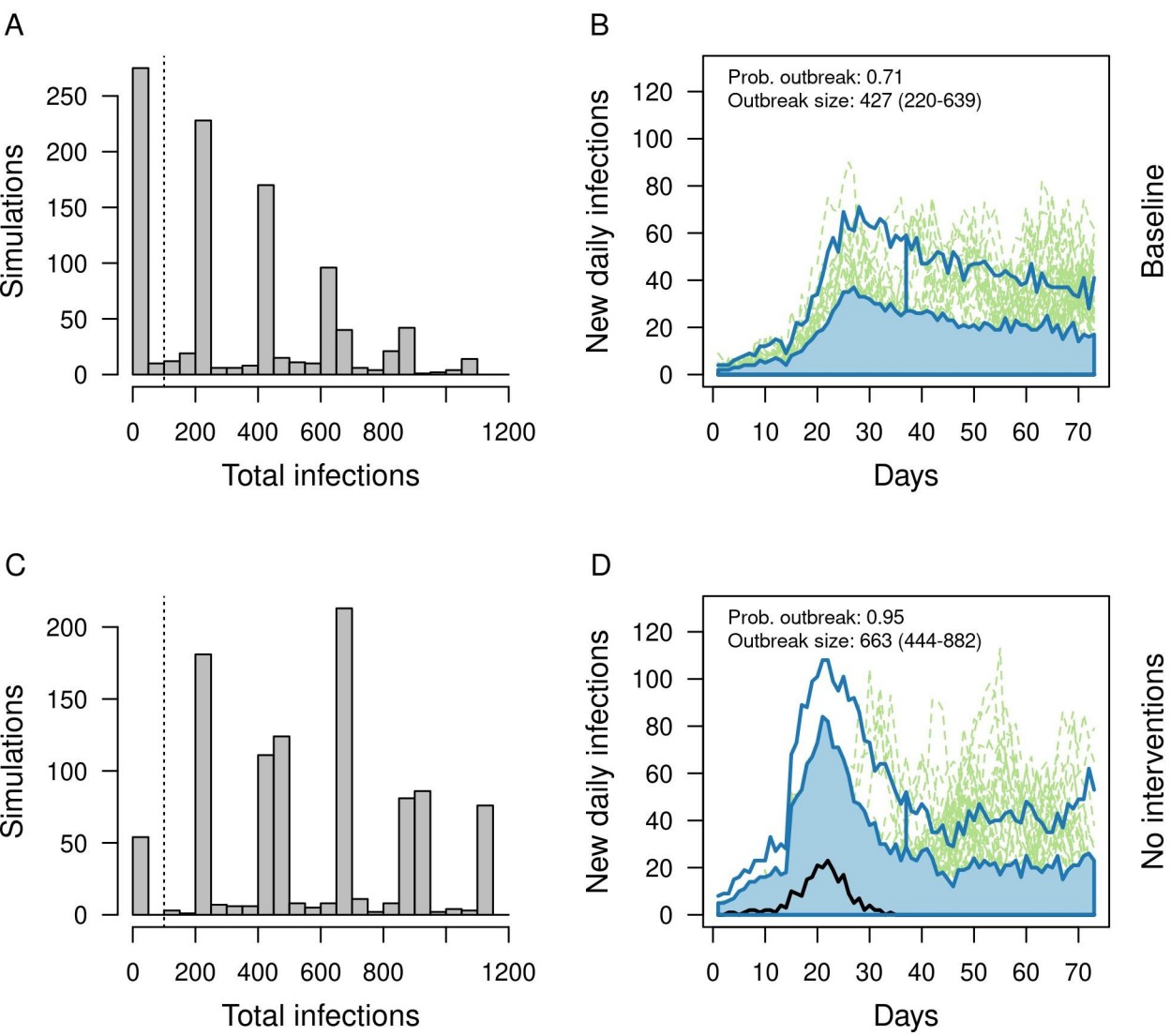

**Fig 3. Outbreaks under the baseline scenario (top) and a scenario with no interventions (bottom).** Left: Distributions of cumulative infections over the 70-day training period across 1,000 replicate simulations. Right: The functional boxplot shows the median estimate (black line), 50% central region (25–75%) (blue band), 1.5 times central region (blue lines), and outliers defined as lines outside of the 1.5 times the central region (dashed green lines). Based on A, we defined an outbreak as 100 or more total infections (vertical dashed lines in A and C). The probability of an outbreak and the median and 25–75% interval of outbreak sizes are printed in B and D.

outbreaks happened in all 1,000 (100%) replicate simulations and had a large magnitude (median: 1,049 infections; 25–75% interval: 863–1,082).

**Arrival testing of trainees.** To isolate the effects of different strategies for arrival testing of trainees, we focused our analysis of testing strategies for trainees on a scenario in which there were no introductions by trainers (Fig 5A and 5B). Compared with no arrival testing of trainees, our baseline scenario of testing on arrival and day 14 reduced the probability of an outbreak from 0.87 (95% confidence interval: 0.85–0.89) to 0.47 (95% CI: 0.44–0.51) (Fig 5A). The second test on day 14 resulted in a modest benefit, with testing on arrival only resulting in an outbreak probability of 0.49 (95% CI: 0.47–0.53). Adding a third test on day seven (median: 0.415; 95% CI: 0.38–0.45) or using an alternative strategy of testing on arrival and days three and five (median: 0.43; 95% CI: 0.39–0.46) resulted in slightly lower outbreak probabilities

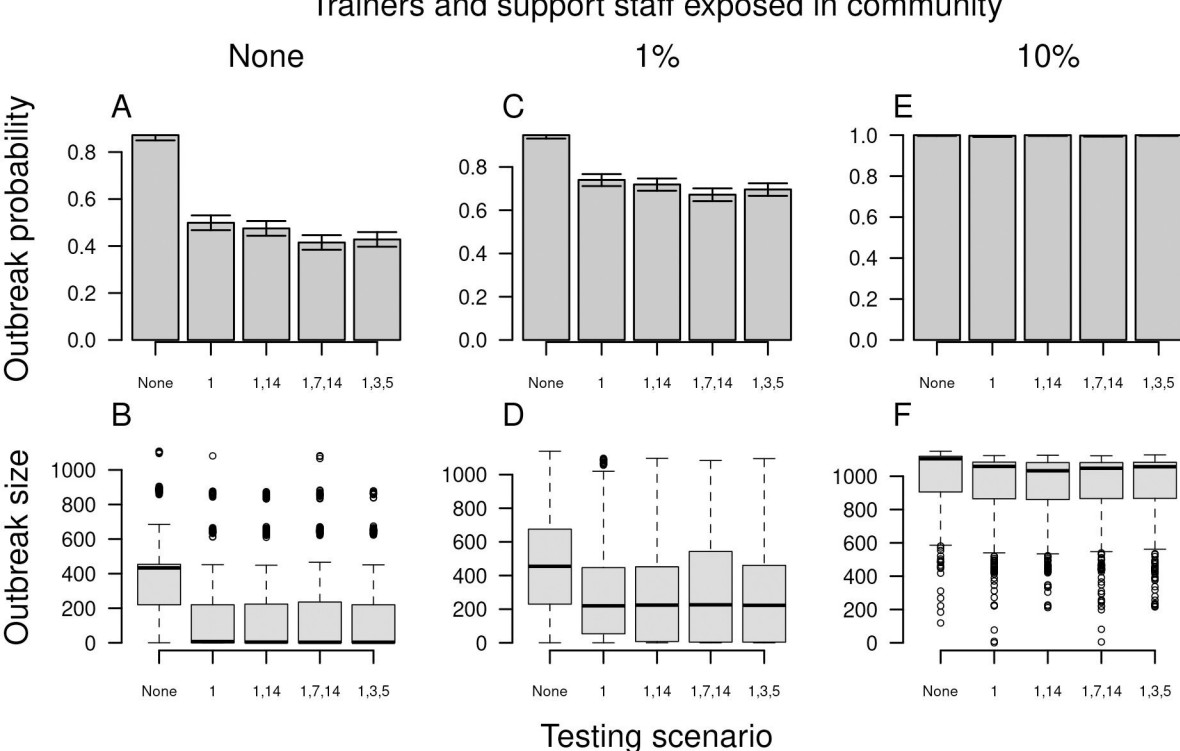

**Fig 4. Outbreaks in basic training as a function of community exposure of trainers and support staff.** From left to right, columns show increases from 0 to 0.01 to 0.10 of the probability that trainers and support staff were exposed to the virus in the community over the course of the 70-day training period. Each panel contains a functional boxplot of the daily incidence of new infections across 1,000 replicate simulations showing the median estimate (black line), 50% central region (25–75%) (blue band), 1.5 times the 50% central region (blue lines), and outliers defined as curves outside of the 1.5 times the central region (dashed green lines). The probability of an outbreak and the median and 25–75% interval of outbreak sizes are printed in each panel.

**Fig 5. Outbreak probability (top) and size (bottom) in basic training as a function of alternative scenarios for testing trainees upon arrival (x-axis).** Testing scenarios are labeled according to the day on which a test was administered to trainees following their arrival. From left to right, columns show increases from 0 to 0.01 to 0.10 of the probability that trainers and support staff were exposed to the virus in the community over the course of the 70-day training period. Error bars for outbreak probability indicate 95% Pearson-Clopper confidence intervals.

than the baseline scenario. Outbreak size was reduced similarly under all scenarios that made use of one or more arrival tests for trainees (Fig 5B). Under a scenario in which trainers and support staff had a 1% chance of community exposure, the relative effects of different strategies for testing trainees were similar, but somewhat less pronounced (Fig 5C and 5D). Under a scenario with 10% community exposure of trainers and support staff, the effects of different testing strategies were minimal, given that introductions by trainers and support staff were the primary driver of outbreaks (Fig 5E and 5F).

**Compliance with face masks and physical distancing.** Across the full range of 0 to 100% compliance with face masks and physical distancing, there was three-fold variation in the probability of an outbreak when there were no introductions by trainers or support staff (Fig 6A). At baseline levels of community exposure to trainers and support staff, this was reduced to less than two-fold variation in outbreak probability (Fig 6C), and to around the same outbreak probability when community exposure to trainers and support staff was high (Fig 6E). The effect of compliance on outbreak size was approximately linear, with reductions being highest under a scenario with high levels of introductions by trainers and support staff, given that outbreaks were so large in that scenario when compliance was zero (Fig 6F). When compliance changed over the course of the training period, outbreak probability was affected minimally (S2A–S2C Fig). At the highest level of community exposure to trainers and support staff, a modest effect of changes in compliance over time could be seen for outbreak size, with higher final compliance reducing outbreak size somewhat (S2F Fig). In addition to outbreak probability and size, high compliance with face masks and physical distancing resulted in outbreaks with lower peak incidence but that were more prolonged (S3 Fig).

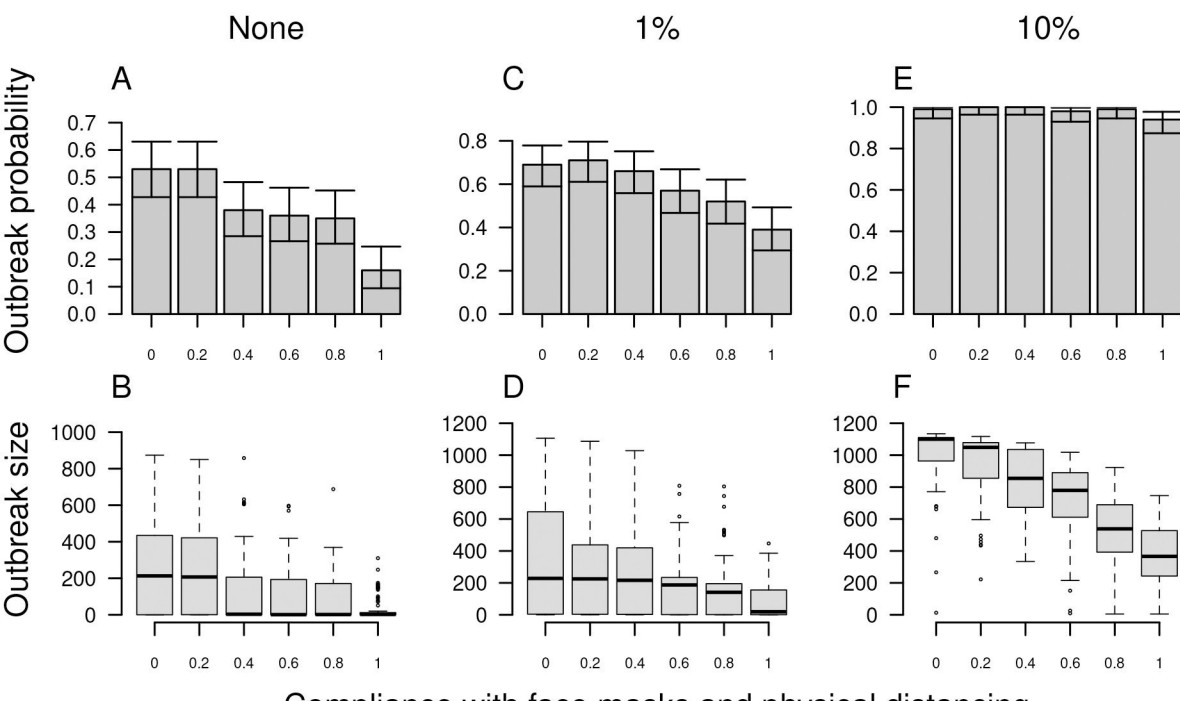

**Fig 6. Outbreak probability (top) and size (bottom) in basic training as a function of compliance with face masks and physical distancing (x-axis).** From left to right, columns show increases from 0 to 0.01 to 0.10 of the probability that trainers and support staff were exposed to the virus in the community over the course of the 70-day training period. Error bars for outbreak probability indicate 95% Pearson-Clopper confidence intervals.

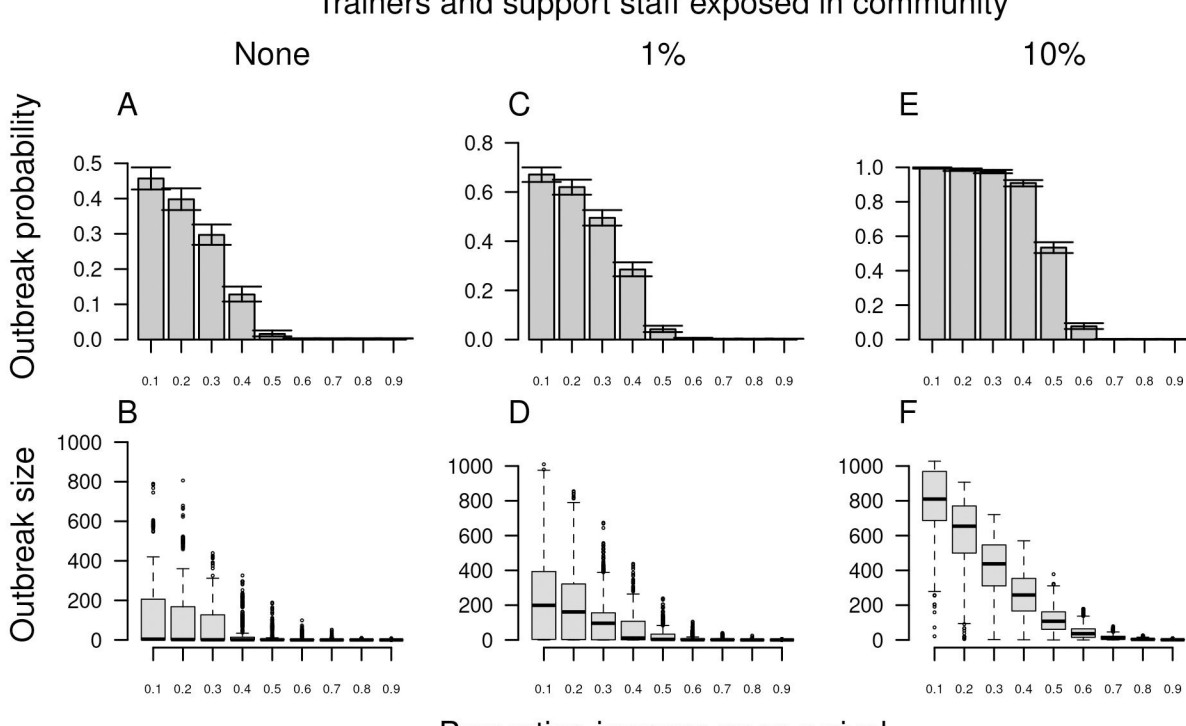

**Fig 7. Outbreak probability (top) and size (bottom) in basic training as a function of the proportion immune upon arrival (x-axis).** From left to right, columns show increases from 0 to 0.01 to 0.10 of the probability that trainers and support staff were exposed to the virus in the community over the course of the 70-day training period. Error bars for outbreak probability indicate 95% Pearson-Clopper confidence intervals.

**Pre-arrival vaccination.** Given the continually changing nature of the landscape of COVID-19 vaccine uptake and effectiveness, we chose to model vaccination in a simple way: by varying the proportion of agents immune at the beginning of the simulation. This corresponds to an all-or-none model of vaccination [30] in which the product of coverage and efficacy against infection equals our parameter for proportion immune. For example, 80% efficacy against infection and 80% coverage would correspond to 64% immune. As the proportion immune in our model increased from 0.10 to 0.90, outbreak probability dropped steeply, with very few outbreaks occurring once the proportion immune at the beginning of the training period reached around 0.40–0.60 (Fig 7A and 7C and 7E). When outbreaks did occur, they were smaller when the proportion immune was higher (Figs 7B and 7D and 7F, and S4). That was particularly so when the rate of virus introduction from trainers and support staff was high, given that immunity reduced the number of companies in which outbreaks occurred. When compliance with face masks and physical distancing was set to zero (S5 Fig), slightly higher levels of immunity were required to achieve the same benefits achieved by lower immunity in the presence of 30% compliance with face masks and physical distancing. In general, for high levels of compliance with facemasks, the levels of immunity required for the probability of an outbreak to be zero was lower (S11 Fig). For instance, assuming high levels of introductions and 100% compliance with facemasks, previous immunity would need to be 40% to result in the probability of an outbreak to be zero. In contrast, at no compliance with facemasks, previous immunity would need to be 70% to achieve the same benefits (S11E Fig).

## Sensitivity analysis

For model parameters not evaluated in our analysis of interventions, we performed a sensitivity analysis to understand how variability in those parameters could affect outbreak probability and size. Under the parameter ranges that we explored, most resulted in outbreak probabilities within 0.2 of baseline and outbreak sizes within 200 infections of baseline (Fig 8).

With respect to outbreak probability, sensitivity was greatest to lower values of the initial prevalence of infection among trainees, as well as the parameters for the generation interval distribution and test sensitivity (Fig 8A). Those parameters all influence the probability that infections among arriving trainees are missed and go on to produce secondary infections. There was also considerable sensitivity to low values of test specificity (Fig 8A). Investigating this further, we found that testing three times (either on days 1, 7, 14 or 1, 3, 5) increased outbreak probability when test specificity was low (S6A and S6C and S6E Fig), whereas those additional tests decreased outbreak probability under baseline test specificity (Fig 5A and 5C and 5E). This suggests that, as a result of lower specificity, additional individuals with false positive test results go on to become infected in group isolation and then return to training, where they contribute to the development of outbreaks. As a potential remedy to this problem, we assessed the impact of testing upon exit from group isolation. Under a scenario with low specificity and no introductions by trainers or support staff, we found that testing reduced outbreak

**Fig 8. Univariate sensitivity analysis.** Changes in outbreak probability (top) and outbreak size (bottom) relative to the baseline scenario are shown by the width of each bar. From left to right, columns show increases from 0 to 0.01 to 0.10 of the probability that trainers and support staff were exposed to the virus in the community over the course of the 70-day training period.

probability from 0.83 (95% CI: 0.81–0.85) to 0.55 (95% CI: 0.52–0.58) (S7 Fig). This brought outbreak probability back within the range expected under higher values of test specificity (median: 0.55; 95% CI: 0.52–0.59). Even when test specificity was at its baseline value to begin with, testing upon exit from group isolation further reduced outbreak probability (S7 Fig). While this form of testing could unnecessarily prolong the time spent by individuals in group isolation who may no longer be infectious, we found that testing upon exit from group isolation did not significantly increase the total person-days in group isolation (S8 Fig, top). In fact, when test specificity was low, testing upon exit from group isolation reduced total person-days in group isolation, given that there were fewer infections who entered group isolation in the first place (S8 Fig, bottom).

The parameters to which outbreak size was sensitive differed according to the extent of community exposure for trainers and support staff. When there were no introductions from trainers or support staff, the greatest sensitivities were to low values of test specificity and longer isolation periods (Fig 8B). As for outbreak probability, this behavior was attributable to false positives becoming infected in group isolation and seeding outbreaks upon return to training, which happened more when specificity was low and a longer isolation period prolonged exposure in group isolation. Likewise, this problem was mitigated by testing upon exit from group isolation (S9 Fig). At our baseline level of community exposure for trainers and support staff, there was moderately high sensitivity to several parameters (Fig 8D). At the highest level of community exposure, sensitivity of outbreak size was greatest to low values of $R_0$ and high compliance with face masks and physical distancing, with smaller outbreak sizes in both cases (Fig 8F).

## Discussion

Calibration of our model to data from two known outbreaks in military basic training settings resulted in a point estimate of initial prevalence among recruits of around 1% at that time, with testing from one of those outbreaks yielding zero positive tests upon arrival and implying one or more false-negative test results. Despite the implication of this result that there should be a steady stream of infections among incoming trainees, our results showed that outbreaks are not an inevitability under these circumstances, with more than half of simulations under our baseline scenario resulting in no outbreak. Accordingly, chance is likely to play a role in why more outbreaks in basic training have not been reported during the pandemic. Higher compliance with face masks and physical distancing than we assumed in our baseline scenario (30%) could also contribute to the prevention of outbreaks in some cases. At the same time, changes in the prevalence of SARS-CoV-2 in communities across the United States [24] are likely to make the risk of undetected introductions by trainees highly dynamic over the course of the pandemic, as they affect the prevalence of infection among trainees. Similar concerns about community transmission for risk of COVID-19 outbreaks in institutional settings have also been raised for K-12 schools [31–35].

Although introductions of SARS-CoV-2 by trainees have been implicated in high-profile outbreaks such as the ones we used to calibrate our model, our analysis predicts that trainers and support staff could play an even greater role in introducing the virus into basic training settings. Whereas trainees enter the training post once, are tested upon arrival, and do not interact with the surrounding community until completion of training, trainers and support staff come and go on a nightly basis over the entire period of training and are not tested unless they present with symptoms. Thus, even though trainees considerably outnumber trainers and support staff, the latter have a much greater chance of becoming infected at some point during the training period and are also more likely to be present on days on which they are maximally

infectious. Trainees, on the other hand, are only likely to transmit appreciably if infected within a few days prior to arrival [22,23]. Our results suggest that the risk of outbreaks in basic training could be reduced considerably if introductions by trainers and support staff could be prevented. In the absence of vaccination, one means of doing so could be to have them remain on post during the training period. Another could be to test them frequently to screen for asymptomatic and presymptomatic infections [36,37]. Once vaccines did become available, a strategy was adopted at one Army basic training post in response to this study whereby trainers and support staff were categorized as Front Line Essential Workers and prioritized for vaccination.

One unique feature of how COVID-19 is managed in basic training that strongly influenced our results is the fact that individuals who test positive are placed into isolation as a group along with others who test positive. In theory, individual versus group isolation should not be of much consequence if everyone in group isolation has already been infected, but in practice this could lead to new infections for individuals who enter group isolation as a result of a false-positive test result. Our results showed that this possibility means that increasing rounds of testing after arrival could come with the downside of producing more false-positive test results and seeding outbreaks once those individuals return to training units. Likewise, our sensitivity analysis showed that seemingly minor imperfections in test specificity can exacerbate this phenomenon. As long as group isolation remains logistically necessary, our results indicate that testing upon exit from group isolation is a promising strategy for mitigating this risk. Importantly, our results also demonstrate that this form of testing appears to be a practical solution, as it does not substantially increase the time that trainees spend in isolation and, under some scenarios, may actually reduce it.

Our calibration resulted in estimates of the basic reproduction number, $R_0$, of 11.3 (95% CrI: 4.9–17.9) and 10.4 (95% CrI: 4.5–17.8) in the two outbreaks used in our calibration. Although the central estimates were most consistent with the data from those outbreaks, we opted for lower-bound estimates given our perception that these outbreaks were not representative of basic training experiences during the pandemic more generally. While even these lower-bound estimates are higher than many $R_0$ estimates in community settings [38,39], it is common for $R_0$ estimates from congregate living settings like basic training to be higher. For example, $R_0$ was estimated to be at least 6.7 for an outbreak on the Diamond Princess cruise ship [40]. Other congregate living settings including homeless shelters [41,42], colleges and universities [1,43], overnight summer camps [44,45], and prisons [46,47] have all experienced high attack rates suggestive of high basic reproduction numbers. A relatively high $R_0$ in basic training makes preventing outbreaks with face masks and physical distancing more difficult, particularly given that trainees sleep in group quarters. It also means that vaccines will need to be highly effective at blocking transmission to prevent outbreaks in basic training settings. Assuming that symptomatic individuals continue to be tested and isolated if positive well into the future, our results suggest that half or more of trainees would need to be fully protected from infection for outbreaks in basic training to be prevented altogether.

Consistent with a long history of research on military medicine translating into benefits for civil society [19], our findings have implications for COVID-19 prevention in institutional settings beyond military basic training. Some of the most visible work modeling COVID-19 in relation to institutional settings has focused on surveillance screening in generic workplace environments [36,37]. An aspect relevant to many institutional settings that generic models neglect is the differential nature of how some classes of individuals interact with the institution. In military basic training, there are two classes: one with a continuous risk of introducing the virus into the institution (trainers and support staff) and another with a one-time risk of doing so (trainees). Two-class structures apply in other institutional settings, as well—e.g., staff and

students in a university, guards and inmates in a prison. In universities, imperfect entry testing of students has been implicated as playing a role in COVID-19 outbreaks early in a new semester [9]. Our work suggests that more than one round of entry testing and individual, rather than group, isolation may be important for mitigating such outbreaks. In prisons, our work suggests that reducing introductions of SARS-CoV-2 by guards is likely to be a critical means of prevention. As in military basic training, measures that are recommended in generic settings (e.g., frequent testing, individual isolation) may not be practical, or even advisable, in prisons. In this way, our work not only offers lessons for those settings, but points to the need for additional work to devise solutions appropriate to them.

One limitation of our analysis is that we did not have detailed information on contact structure within training units. In the absence of this information, we made the simplifying assumption that everyone within a training unit had equal contact with everyone else. There are also details about sex segregation at certain stages of the basic training process that we did not consider and could affect contact patterns. Because contact heterogeneity is thought to be a primary driver of individual heterogeneity in transmission [48], our model was not well-suited to addressing the potential role of superspreading in the basic training setting. In addition, there is also uncertainty regarding the extent and nature of contacts among trainees, trainers, and support staff. Because outbreak probability was strongly influenced by the probability of introductions by trainers and support staff, studies of the relative strength of trainee-trainee and trainer-trainee contacts could be important for refining understanding of outbreak risk in basic training settings. We also did not evaluate the potential impact of contact tracing in this setting. While contact tracing has proven effective in other settings [49], it may be difficult to implement effectively in this setting because the frequency and nature of contacts within a relatively large group make standard definitions of close contacts uninformative [50]. There are uncertainties about some of our parameter values, such as mask effectiveness, baseline immunity, and testing accuracy. We addressed these uncertainties through a sensitivity analysis, which showed that outbreak probabilities remained similar to our baseline scenario under a range of parameter values. The parameters that did significantly impact outbreak probability (community exposure of trainers and support staff, pre-arrival immunity) are likely to vary over the course of the pandemic, with our estimates offering intuition about how outbreak probability and size could change as a result.

In conclusion, our results show that military basic training is a unique setting that requires customized strategies for preventing COVID-19 outbreaks. Specifically, we show that while testing of trainees upon arrival is important, frequent testing of trainers and support staff who interact with trainees may be even more important. Likewise, it draws attention to the high priority that should be placed on trainers and support staff for vaccination, which may be a more actionable recommendation based on this work. Unlike other settings, our results show that testing of trainees that is too frequent could come with the drawback of increasing the risk of an outbreak. This counterintuitive result is a consequence of the fact that false positives could result in susceptible trainees becoming infected in group isolation and then seeding an outbreak in their training unit upon release from isolation. Like other settings, our results suggest that compliance with face masks and physical distancing is important and that a transmission-blocking vaccine could be effective at preventing outbreaks. At the same time, the relatively high values of $R_0$ that we estimated from two outbreaks in basic training settings imply that these interventions will be less impactful in basic training than in community settings. In the event of other emerging respiratory viruses in the future, our model could serve as a starting point for exploring the possible impacts of such a virus and how best to control it to maintain basic training operations.

## Methods

### Model description

We developed an agent-based model of SARS-CoV-2 transmission in a single cohort of trainees, their trainers, and associated support staff at a single U.S. Army training post, based on informed hypothetical assumptions from the operations in this setting. In reality, new cohorts enter a training post on a weekly basis. Although there is some possibility for an outbreak in one cohort to spill over into another cohort, such outbreaks are likely to be mostly independent of one another given limited contact among trainees and trainers from different weeks' cohorts. As such, we viewed a model of a single cohort as sufficient to inform on the effects of various prevention efforts, which was our primary goal in this study. Because of the rapid time-scale of outbreaks in this setting, we modeled all processes on a daily time step. Below, the model is described in general terms, with parameter values provided in Table 1.

The model was implemented in the R programming language version 4.0 [51] using the packages *scam* [52] and *igraph* [53]. A single realization of the model takes around a second to execute on a personal computer using Linux (Fedora 34) on a single thread. The simulation analyses were performed using the supercomputing infrastructure of the Center for Research Computing at Notre Dame (https://crc.nd.edu). Different realizations of the model were simulated in parallel, but each simulation was performed on a single-thread computing node. All code used in this analysis is available at https://github.com/confunguido/prioritizing_interventions_basic_training.

**Structuring of agents and their contacts.** Our model included a total of 1,200 trainees, 40 trainers, and 60 support staff (Fig 1). Trainees arrived over a three-day window and proceeded to one of 20 cocoons of 60 recruits each. After 14 days, five companies of 240 recruits each were formed by pooling together four cocoons. Pooling of individuals into cocoons and cocoons into companies was done randomly in the model and not with respect to vaccination status, gender, or any other factor. Trainees remained in their company for an additional 56 days until training was completed. Throughout the 70-day training period, trainees were in contact with other trainees in their unit (initial cocoon and then company) and with trainers assigned to their unit: two trainers for each cocoon and eight for each company. Trainees also came into contact with a set of 60 support staff, which includes staff providing support for dining, shooting ranges, equipment, and first aid. In the event that trainees tested positive for SARS-CoV-2, they were separated from their unit and placed in the sick bay, where they had contact with everyone else in the sick bay. Trainers and support staff who tested positive isolated at home, meaning that they had no contact with any other agents in the model during that time.

**SARS-CoV-2 infection and transmission.** Introduction of SARS-CoV-2 into the cohort occurred by two means: through trainees upon arrival or through trainers or support staff at any point over the training period. Whereas trainees are restricted to the training setting once they arrive, trainers and support staff go home at night and engage in day-to-day activities in the community in their time away from work. We simulated initial infections among trainees according to a binomial random variable based on their initial prevalence of infection and simulated the timing of any initially infected trainee's infection as a uniform random variable between one and 39 days prior to arrival. This period was chosen based on the period in which test sensitivity was assumed to exceed zero under our model, which also encompasses the period of infectiousness. Because trainers and support staff could become infected at any point during the training period in the community surrounding the training post, we simulated community-acquired infections with a daily probability consistent with a given infection attack rate over the 70-day period of training. We chose values for this probability consistent

with infection attack rates over a 70-day period spanning a range of estimates by Pei et al. [24] from the four states with U.S. Army basic training posts from May through July, 2020 (S10 Fig). At the same time, trainers and support staff could also become infected within the training environment and were subject to the same model parameters as trainees pertaining to that environment.

The course of each agent's infection was defined on a daily basis relative to their day of exposure. In terms of infectiousness, the probability that an infected agent transmits to a susceptible contact on a given day of infection is proportional to the value of the generation interval distribution for that day, which we modeled with a Weibull distribution [27]. Because our model operates on a daily time step, we used a discretized version of this distribution wherein the probability of an interval of length $t$ was $p(t) = F(t+1) - F(t)$, where $F(t)$ is the distribution function. The magnitude of infectiousness was captured by a parameter that was multiplied with the generation interval distribution value on a given day of infection, resulting in a daily probability of transmitting to a given susceptible contact. The sum of those daily probabilities across all days of infection multiplied by the average number of contacts was equivalent to $R_0$. Only a subset of agents develop symptoms, with that outcome determined by a Bernoulli trial for each infected agent. For those who do, symptoms manifest according to an incubation period drawn from a discretized gamma distribution [22], and symptoms conclude a number of days later drawn from a Poisson distribution [26]. For agents who remain asymptomatic, their probability of transmitting to one of their contacts is lower than for their symptomatic counterparts.

We assumed that a small proportion of individuals in the model were previously infected prior to arrival of trainees, consistent with the timing of reported outbreaks at two U.S. Army training posts in spring 2020. Given that those outbreaks were reported on May 31, 2020 and were based on testing on days 18 and 22 of training [17,54], we assumed that those trainees likely arrived during the week of May 3, 2020. Thus, estimates of cumulative incidence of infection prior to that time should provide a reasonable approximation of previous exposure and immunity. Based on estimates from a study [24] that modeled cumulative infections in the U.S. population over the course of the epidemic, a median estimate of immunity among trainees as of May 3, 2020 was 2.6% (95% CrI: 1.8–3.3%). These estimates are national averages of state-level estimates weighted by state-level Army recruitment rates [55]. We used the median estimate in the model calibration and baseline scenario, and we explored the lower and upper values in a sensitivity analysis.

**Interventions.**   The primary means of preventing transmission in the model involved testing for active infection and isolating test-positives. In our baseline scenario, trainees were tested upon arrival and 14 days later, as well as any time they developed symptoms. Trainers and support staff were also tested if they displayed symptoms. There was a modest delay of one day between the time that a test was administered and when results were available. Individuals continued with their normal activities while awaiting test results, entering isolation in the event of a positive result and remaining there for ten days [29].

Test sensitivity varied by day of infection according to a piecewise model of daily test sensitivity proposed by Grassly et al. [37]. In days one through six after infection, daily test sensitivity is proportional to daily infectiousness. In days seven and after, daily test sensitivity declines according to a curve estimated with a generalized additive model by Wikramaratna et al. [56]. To allow for flexibility in the magnitude of sensitivity, we multiplied the curve for daily test sensitivity by a scalar such that an average of daily test sensitivity weighted by the incubation period distribution equaled a parameter for overall test sensitivity. This approach to calculating a weighted average of daily test sensitivity resulted in the sensitivity of tests applied to

individuals presenting with symptoms being equal to the parameter for overall test sensitivity, on average. For specificity, we assumed a constant value.

We chose values of overall test sensitivity and specificity based on data from an analysis of more than 800 individuals tested two to three times each on the same day with a combination of PCR tests of nasal swab specimens, PCR tests of saliva specimens, and antigen tests of nasal swab specimens [28]. A Bayesian latent class analysis of those data obviated the need to define any one of those tests as a gold standard by simultaneously accounting for imperfect sensitivity and specificity of each test. The majority of individuals in that data set were college students tested for surveillance purposes, meaning that their detectability of infection should be very similar to surveillance testing in a military basic training population. We used median values of estimates for PCR tests of nasal swab specimens, which were 0.859 (95% CrI: 0.547–0.994) for sensitivity and 0.998 (95% CrI: 0.992–0.999) for specificity. These values of sensitivity were similar to estimates from a meta-analysis of 16 published studies (median: 0.848; 95% CrI: 0.768–0.924) [57]. We were unable to find other studies on the specificity of clinical testing with PCR tests, but similar ranges were found in a meta-analysis that evaluated data from 2004–2019 on 43 studies of PCR tests for other RNA viruses [58].

In addition to testing, we assumed that agents made use of face masks and physical distancing, when possible, to reduce transmission. These interventions impacted transmission by reducing the probability of transmission between an infectious agent and one of their contacts proportional to the probability that either or both agents were in compliance with face-mask and physical-distancing guidelines at the time of contact and the per-contact reduction in the probability of transmission from these interventions.

## Model calibration

We calibrated the model to two known outbreaks in U.S. Army training posts: Fort Benning (FB) [54] and Fort Leonard Wood (FLW) [17]. In both cases, we made use of reports of positive tests upon arrival (FB: 4/640; FLW: 0/500) and following an initial period of group quarantine (FB: 142/636 on day 22; FLW: 70/500 on day 18). We used a two-step approach that leveraged the information at these time points in a sequential manner. Because no information about infections among trainers or support staff were provided in these reports, we limited the calibration to infections among trainees only. We performed this calibration procedure separately on the data from FB and FLW.

In the first step, we used data on the number of positive tests upon arrival, $Positive_{Arrival}$, to inform an initial estimate of the prevalence of infection among trainees upon arrival, $p$. According to our assumptions about test sensitivity as a function of day of infection, 99% of positive tests should have resulted from individuals infected within 39 days of arrival, assuming a constant rate of infection over that period. Given the average test sensitivity, $Se$, over this 39-day period and the test specificity, $Sp$, we defined the likelihood of $p$ according to

$$L(p|Positive_{Arrival}) = Binomial(Positive_{Arrival}|Tested_{Arrival}, p \times Se + (1-p) \times (1-Sp)),$$

where $p$ x $Se$ + $(1—p)$ x $(1—Sp)$ is the probability of a trainee testing positive when accounting for imperfect test performance. We defined the posterior probability density of $p$ as

$$Pr(p|Positive_{Arrival}) = \frac{L(p|Positive_{Arrival})}{\int_0^1 L(p|Positive_{Arrival})dp},$$

which assumes a uniform prior on $p$.

In the second step, we used approximate Bayesian computation to select combinations of $p$ and $R_0$ that were consistent with data on the number of positive tests following group

quarantine, $Positive_{Later}$. The posterior distribution of $p$ from the first step served as a prior distribution of $p$ in the second step. The initial set of particles were comprised of draws of $p$ from the estimated distribution combined with draws of $R_0$ from a normal prior distribution (mean 9.6; 95% CI: 6.9–12.4) obtained from three different estimates from the literature [40,44,59]. Each of 200,000 of these particles was used to simulate forward under the model one time until day 22 at FB and day 18 at FLW. In these simulations, the timing of infection of trainees infected upon arrival was drawn uniformly from one to 39 days prior to arrival, given that this was the period of time over which test sensitivity was allowed to exceed zero under our model. For a given particle $i$, $Tested_{Later}$ trainees were tested (FB: 636; FLW: 500), and the number positive, $Positive^i_{Later}$, was recorded. Particles for which $Positive^i_{Later}$ equaled the observed $Positive_{Later}$ (FB: 142; FLW: 70) were retained, the set of which comprised our approximation of the posterior distribution of $p$ and $R_0$ for each of FB and FLW. Given that the observed outbreaks were likely exceptional events rather than common occurrences, we focused our baseline scenario on a value of $R_0$ equal to the average of the lower bounds of the $R_0$ estimates from FB and FLW.

## Analyses

**Model behavior under baseline scenario.**   Following calibration of the model, we added average values of initial prevalence and $R_0$ to the list of baseline parameter assumptions in Table 1. Under this baseline scenario, we performed 1,000 replicate simulations with the hypothetical cohort portrayed in Fig 1, examining the time course of the outbreak across replicates, the probability of a large outbreak, and the size of a large outbreak, if one occurred. We evaluated these same three model outputs under varying levels of four factors that could be altered by interventions.

**Reducing introductions by trainers and support staff.**   On the one hand, community exposure of trainers and staff could go up or down depending on the prevalence of SARS-CoV-2 in the community at any given time. On the other hand, introductions from this source could potentially be reduced by regularly testing trainers and support staff [36] or by having them remain on base for the duration of the training period. To understand the implications of different rates of introduction by trainers and support staff, we performed simulations under three different levels of introduction (0, 1%, 10%), defined as the proportion of trainers and support staff infected in the community over the course of the 70-day training period. This broad range of variation should cover the full range of possible exposure during this time window, including periods of low and high levels of community transmission.

**Arrival testing of recruits.**   In addition to the baseline scenario of PCR tests on days 0 and 14, we considered a scenario without the test on day 14, a scenario with an additional test on day 7, a scenario with tests on arrival and days 3 and 5, and a scenario with no arrival testing.

**Compliance with face masks and physical distancing.**   Our default assumption was that compliance with face masks and physical distancing was relatively low (30%) due to the physically intense nature of training and the fact that training entails large groups of people spending prolonged periods of time together. Given uncertainty about the appropriateness of our baseline assumption of 30% and the potential for compliance with these measures to be either disregarded completely or enforced strictly, we explored scenarios in which compliance ranged from 0 to 100%. We also explored scenarios in which compliance began at the baseline value of 30% and either decreased or increased linearly over the course of the training period (to 10%, 20%, 40%, or 50%). The purpose of these scenarios was to understand the possible impact of behavioral change over the course of the training period, were trainees to relax their precautions or heighten them.

**Pre-arrival vaccination.**   Immunity among trainees, trainers, and support staff varies naturally depending on the history of the epidemic in communities that these individuals come

from and the time in the epidemic when they arrive. Similarly, vaccination coverage varies across communities, as well. For both forms of immunity, waning can compound variability in protection against infection at different points of time in different groups of people. To cover this wide range of possible scenarios, we varied the proportion immune upon arrival from 0 to 90% in increments of 10%.

**Sensitivity analysis.** To understand sensitivities of the model's behavior to parameters not explored in the intervention analyses, we conducted a univariate sensitivity analysis for all other model parameters. These parameters, and the alternative low and high values that we explored, are listed in Table 1. For each alternative parameterization, we ran 1,000 simulations and calculated the probability of an outbreak and the size of one, if it occurred.

Wherever possible, we selected high and low values based on upper and lower bounds of 95% confidence or credible intervals from studies that estimated those parameters. Our reasoning for doing so was to convey the extent to which model outputs might change within a plausible range of uncertainty about a given parameter. At the same time, we note that a limitation of this approach is that it does not convey the full uncertainty in model outputs attributable to parameter uncertainty, which would require a fuller accounting of joint uncertainty across all model parameters (e.g., as in [60]). Accordingly, this analysis is intended to aid in the building of intuition of decision makers rather than to provide quantitative projections.

For three parameters, we chose values based on our judgement about what constituted reasonable ranges, due to difficulty in identifying reliable descriptions of uncertainty for those parameters. The first of those parameters was $R_0$, which had extremely wide ranges of uncertainty in our model calibration that likely exceed the true range of uncertainty about this parameter. The second of those parameters was the probability of community exposure to trainers and support staff, values of which were loosely based on estimates of time-varying prevalence of infection in the four states with U.S. Army training posts in May 2020 (S10 Fig). The third of those parameters was the relative infectiousness of asymptomatics, which we perceive to be generally viewed as somewhat less than that of symptomatic infections but not to a great extent [61].

## Disclaimer

Material has been reviewed by the Walter Reed Army Institute of Research and the U.S. Army Training and Doctrine Command. There is no objection to its presentation and/or publication. The opinions or assertions contained herein are the private views of the authors, and are not to be construed as official, or as reflecting true views of the Department of the Army or the Department of Defense.

## Supporting information

**S1 Fig. Simulation results used for the model calibration.** We simulated 200,000 replicate outbreaks for values of $R_0$ evenly spaced between 2 and 20. The horizontal line shows the observed data.
(TIF)

**S2 Fig. Outbreak probability (top) and size (bottom) in basic training as a function of final proportion of compliance with face masks and physical distancing (x-axis).** The starting proportion of compliance was set to the baseline value of 0.3, which linearly increased or decreased over time to its final value by the end of the training period. From left to right, columns show increases from 0 to 0.01 to 0.10 of the probability that trainers and support staff were exposed to the virus in the community over the course of the 70-day training period.

Error bars for outbreak probability indicate 95% Pearson-Clopper confidence intervals.
(TIF)

**S3 Fig. Outbreaks in basic training as a function of compliance with face masks and physical distancing.** From left to right, columns show increases in the proportion of time that individuals comply with face masks and physical distancing. From top to bottom, rows show increases from 0 to 0.01 to 0.10 of the probability that trainers and support staff were exposed to the virus in the community over the course of the 70-day training period. Each panel shows a functional boxplot of the daily incidence of new infections across 1,000 replicate simulations, showing the median estimate (black line), 50% central region (25–75%)(blue area), 1.5 times the central region (blue lines), and outliers defined as curves outside of the 1.5 times the central region (dashed green lines).
(TIF)

**S4 Fig. Outbreaks in basic training as a function of the proportion immune upon arrival.** From left to right, columns show increases in the proportion immune upon arrival. From top to bottom, rows show increases from 0 to 0.01 to 0.10 of the probability that trainers and support staff were exposed to the virus in the community over the course of the 70-day training period. Each panel shows a functional boxplot of the daily incidence of new infections across 1,000 replicate simulations, showing the median estimate (black line), 50% central region (25–75%)(blue area), 1.5 times the central region (blue lines), and outliers defined as curves outside of the 1.5 times the central region (dashed green lines).
(TIF)

**S5 Fig. Outbreak probability (top) and size (bottom) in basic training as a function of the proportion immune upon arrival (x-axis) when there is zero compliance with face masks and physical distancing.** From left to right, columns show increases from 0 to 0.01 to 0.10 of the probability that trainers and support staff were exposed to the virus in the community over the course of the 70-day training period. Error bars for outbreak probability indicate 95% Pearson-Clopper confidence intervals.
(TIF)

**S6 Fig. Outbreak probability (top) and size (bottom) in basic training as a function of alternative scenarios for testing trainees upon arrival (x-axis) when test specificity is low (0.992).** Testing scenarios are labeled according to the day on which a test was administered to trainees following their arrival. From left to right, columns show increases from 0 to 0.01 to 0.10 of the probability that trainers and support staff were exposed to the virus in the community over the course of the 70-day training period. Error bars for outbreak probability indicate 95% Pearson-Clopper confidence intervals.
(TIF)

**S7 Fig. Outbreak probability as a function of testing upon exit from group isolation (x-axis).** Rows show results for different values of test specificity. From left to right, columns show increases from 0 to 0.01 to 0.10 of the probability that trainers and support staff were exposed to the virus in the community over the course of the 70-day training period. Error bars indicate 95% Pearson-Clopper confidence intervals.
(TIF)

**S8 Fig. Total person-days in group isolation as a function of testing upon exit from group isolation (x-axis).** Rows show results for different values of test specificity. From left to right, columns show increases from 0 to 0.01 to 0.10 of the probability that trainers and support staff were exposed to the virus in the community over the course of the 70-day training period.
(TIF)

**S9 Fig. Outbreak size as a function of testing upon exit from group isolation (x-axis).** Rows show results for different values of test specificity. From left to right, columns show increases from 0 to 0.01 to 0.10 of the probability that trainers and support staff were exposed to the virus in the community over the course of the 70-day training period.
(TIF)

**S10 Fig. Community exposure over time in states with U.S. Army basic training posts.** This measure of exposure is defined as infection attack rate over a 70-day period commencing on the date indicated on the x-axis in 2020, as estimated by Pei et al. (24). Solid lines show medians, and bands show 95% credible intervals.
(TIFF)

**S11 Fig. Outbreak probability (top) and size (bottom) in basic training as a function of the proportion immune upon arrival (x-axis) and compliance with facemasks and physical distancing (colors).** From left to right, columns show increases from 0 to 0.01 to 0.10 of the probability that trainers and support staff were exposed to the virus in the community over the course of the 70-day training period. Each point reflects a proportion or median across 1,000 replicate simulations.
(TIFF)

## Author Contributions

**Conceptualization:** T. Alex Perkins, Simon D. Pollett, Paul T. Scott.

**Data curation:** Guido España.

**Formal analysis:** Guido España.

**Investigation:** Guido España, T. Alex Perkins, Simon D. Pollett, Morgan E. Smith, Sean M. Moore, Paul T. Scott.

**Methodology:** Guido España, T. Alex Perkins, Sean M. Moore.

**Project administration:** T. Alex Perkins.

**Resources:** Paul O. Kwon, Tara L. Hall, Milford H. Beagle, Jr., Clinton K. Murray, Shilpa Hakre, Sheila A. Peel, Kayvon Modjarrad, Paul T. Scott.

**Software:** Guido España, T. Alex Perkins, Morgan E. Smith.

**Supervision:** T. Alex Perkins, Paul T. Scott.

**Validation:** Guido España, Paul T. Scott.

**Visualization:** Guido España.

**Writing – original draft:** T. Alex Perkins.

**Writing – review & editing:** Guido España, Simon D. Pollett, Morgan E. Smith, Sean M. Moore, Paul O. Kwon, Tara L. Hall, Milford H. Beagle, Jr., Clinton K. Murray, Shilpa Hakre, Sheila A. Peel, Kayvon Modjarrad, Paul T. Scott.

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
