## [Decision Letter · Decision Letter 0]

9 May 2022

Dear Dr. Perkins,

Thank you very much for submitting your manuscript "Prioritizing interventions for preventing COVID-19 outbreaks in military basic training" for consideration at PLOS Computational Biology. As with all papers reviewed by the journal, your manuscript was reviewed by members of the editorial board and by several independent reviewers. The reviewers appreciated the attention to an important topic. Based on the reviews, we are likely to accept this manuscript for publication, providing that you modify the manuscript according to the review recommendations.

This manuscript has already gone through extensive peer review at PNAS, which was provided along with the submission. Given the careful attention paid to what are quite thorough reviewer comments, it's not a surprise that the reviews here have only minor comments. I encourage the authors to take careful note of the comments directed towards further clarifying the work.

Sincerely,

Samuel V. Scarpino

Associate Editor

PLOS Computational Biology

Virginia Pitzer

Deputy Editor-in-Chief

PLOS Computational Biology

[LINK]

This manuscript has already gone through extensive peer review at PNAS, which was provided along with the submission. Given the careful attention paid to what are quite thorough reviewer comments, it's not a surprise that the reviews here have only minor comments. I encourage the authors to take careful note of the comments directed towards further clarifying the work.

Reviewer's Responses to Questions

**Comments to the Authors:**

Reviewer #1: Please see the attached document.

Reviewer #2: The authors model COVID outbreak in a military basic training setting. They estimate a parameter space based on the outbreaks at Forth Benning and Fort Leonard Wood, and with the findings from several published studies (barring the relative infectiousness of asymptomatics). They construct a baseline simulation and compare its results to simulations which vary the parameter space to test the effects of 1) reducing introductions by trainers and support staff, 2) increasing testing of trainees, 3) increasing compliance with wearing face masks and physical distancing, and 4) increasing immunity among trainees through pre-arrival vaccination. The results of the simulations are very informative but not very surprising. However, I think this paper remains a quality addition to the growing library of COVID related research. I have several minor concerns regarding the methodology, and a few concerns about knowledge translation.

In the last paragraph of your introduction, consider changing “That informed the model’s assumptions ...” to an alternative such as “The calibrations informed the model’s assumptions ...” to minimize ambiguity.

In the last paragraph of the introduction you first mention the effect of introductions by trainers and support staff, however I could not find a passage where you outline what introductions entail. Introductions vary between cultures, and I imagine between civilian and military settings. A followup question is whether there are intermediate levels of introduction.

I thought figure 1 was a great pictorial representation of your model, but I would suggest a possible improvement. The transitions and contact column could be represented as a state-diagram/flowchart. This alternative representation would be appealing/recognizable to biologists (e.g. life history cycles) and computer scientists, alike, and might improve knowledge translation.

In your results, first paragraph under “model calibration”, you first reference figure 2. There is no mention within the paragraph or the figure caption for why the outliers (dashed green lines) are not visible – are the outliers superimposed or negligible?

In your results, second paragraph under “model behaviour under baseline scenario” you say, “According to this definition, 71% of simulations resulted in an outbreak ...”, and shortly after “... the probability of an outbreak increased to 0.95 ...”. I would suggest using one representation of probability, rather than two, to minimize ambiguity.

In your results, figure 8, the y-axis labels should either be written out entirely or explained in the figure caption.

In your methods, first paragraph under “model description” you say, “... based on hypothetical assumptions provided by an author (PTS) familiar with operations in this setting”. Is it necessary to cite the author specifically because the information is not available publicly? Otherwise, it seems redundant to say that your research is contingent on its authors’ knowledge, in which case I would suggest an alternative like “based on informed hypothetical assumptions from the operations in this setting.”.

In your methods, first paragraph under “model description”, you mention that you use the R programming language. You should include the version of R you used and any relevant packages. As well, as cite the literature associated with those packages (if not base R). You also say a single realization of the model takes around a second to execute on a personal computer, but you do not specify whether this is on single or multiple threads, the operating system, or the CPU you used, all which have a bearing on speed, especially while running multiple instances. I assume the amount of RAM required is negligible.

In your methods, under “Structuring of agents and their contacts” you say “Trainees arrived over a three-day window and proceeded to one of 20 cocoons of 60 recruits each. After 14 days, five companies of 240 recruits each were formed by pooling together four cocoons.”. You do not mention whether there are any considerations on how recruits are organized into their cocoons, or how cocoons are organized into their companies, in silico or in reality. For example, are trainees/cocoons randomly selected? I would also ask the following hypothetical: if the probability of infection was known for all individuals, how would you distribute them in to cocoons and companies in order to minimize overall transmission?

In your methods, under “Model calibration”, in the third paragraph (following your formulation of the likelihood of p), you say, “The initial set of particles were comprised of draws of p from the estimated distribution combined with draws of R0 from a normal prior distribution ...”. A normal distribution can take-on negative values, whereas the R0 cannot. I am curious as to why you did not choose something like a lognormal distribution? For example lognormal priors have been used to estimate R0 directly (Purkayastha, 2021; https://doi.org/10.1186/s12879-021-06077-9) and indirectly (Mbuvha, 2020; https://doi.org/10.1371/journal.pone.0237126). Some research has also explored using generalized gamma distributions, where a lognormal distribution is a special case.

In your analyses, under “Reducing introductions by trainers and support staff” you do not state your methodology as compared to the “Pre-arrival vaccination” section where you present the same ‘one-hand-other-hand’ intro, and then later state exactly how you vary your parameters. You also state how you vary your parameters in other nearby sections, just not in "Introductions by trainers and support staff”.

I would like to add to one of your previously addressed comments on isolation before arrival. You say the army does not have control over trainees before arrival and could therefore not rely on a voluntary isolation period of two weeks. Canada has set this very protocol for its recruits, and I would like to think it's due to its efficacy. Yes, you can not expect perfect compliance, but partial compliance must still be effective? I do not know if there is data published on the effect of isolation before arrival for the Canadian military, but it is certainly something worth looking into.

You do not finish your discussion with a definitive statement on how you think your model will inform decision-making in a military setting. You also state in your previous set of reviewers comments that you have planned a companion piece to express the “real-wold benefits of [your] research”. I would suggest that you hint at this future work somewhere in your conclusions.

**Have the authors made all data and (if applicable) computational code underlying the findings in their manuscript fully available?**

Reviewer #1: Yes

Reviewer #2: Yes

PLOS authors have the option to publish the peer review history of their article (what does this mean?). If published, this will include your full peer review and any attached files.

Reviewer #1: No

Reviewer #2: **Yes: **Winston Campeau

Figure Files:

Data Requirements:

Reproducibility:

References:

---

## [Editor Report · Decision Letter 1]

12 Aug 2022

Dear Dr. Perkins,

We are pleased to inform you that your manuscript 'Prioritizing interventions for preventing COVID-19 outbreaks in military basic training' has been provisionally accepted for publication in PLOS Computational Biology.

Best regards,

Samuel V. Scarpino

Associate Editor

PLOS Computational Biology

Virginia Pitzer

Deputy Editor-in-Chief

PLOS Computational Biology

---

## [Editor Report · Acceptance letter]

20 Sep 2022

PCOMPBIOL-D-22-00439R1 

Prioritizing interventions for preventing COVID-19 outbreaks in military basic training

Dear Dr Perkins,

I am pleased to inform you that your manuscript has been formally accepted for publication in PLOS Computational Biology. Your manuscript is now with our production department and you will be notified of the publication date in due course.

With kind regards,

Livia Horvath
